# Frequency and Clinical Review of the Aberrant Obturator Artery: A Cadaveric Study

**DOI:** 10.3390/diagnostics10080546

**Published:** 2020-07-30

**Authors:** Guinevere Granite, Keiko Meshida, Gary Wind

**Affiliations:** 1Department of Surgery, Uniformed Services University of the Health Sciences, Bethesda, MD 20814, USA; gary.wind@usuhs.edu; 2Department of Anatomy, College of Medicine, Howard University, Washington, DC 20059, USA; keiko.meshida@howard.edu

**Keywords:** aberrant obturator artery, internal iliac branching variations, external iliac branching variations, anatomical variations

## Abstract

The occurrence of an aberrant obturator artery is common in human anatomy. Detailed knowledge of this anatomical variation is important for the outcome of pelvic and groin surgeries requiring appropriate ligation. Familiarity with the occurrence of an aberrant obturator artery is equally important for instructors teaching pelvic anatomy to students. Case studies highlighting this vascular variation provide anatomical instructors and surgeons with accurate information on how to identify such variants and their prevalence. Seven out of eighteen individuals studied (38.9%) exhibited an aberrant obturator artery, with two of those individuals presenting with bilateral aberrant obturator arteries (11.1%). Six of these individuals had an aberrant obturator artery that originated from the deep inferior epigastric artery (33.3%). One individual had an aberrant obturator artery that originated directly from the external iliac artery (5.6%).

## 1. Introduction

Knowledge of the vascular anatomy and its possible variations is essential to performing embolization and revascularization procedures in the human pelvis [1]. The obturator artery (OA), a standard branch of the anterior division of the internal iliac artery (IIA), has the greatest frequency of variation among the IIA branches [2,3,4,5,6,7,8,9,10,11,12]. Understanding the possible OA origin variations is important while performing pelvic and groin surgeries requiring appropriate ligation. In instances of acute pelvic or groin trauma, such varying origins may be a significant source of persistent hemorrhages that are difficult to manage [10,11,13,14,15,16,17,18,19,20,21,22,23,24,25]. Thus, OA origin variations are an important anatomical topic for a range of medical fields as varied as gynecology, orthopedics, and urology [25,26]. 

The abdominal aorta divides into the right and left common iliac arteries, which further subdivide into the external iliac artery (EIA) and the internal iliac artery (IIA) on each side [11,13,27,28]. The EIA mainly supplies the lower limbs. The IIA usually supplies the pelvis, perineum, and gluteal regions with common anatomical variations [1,11,23,26,28]. Typically, four IIA branches occur in the male, while five occur in the female (see Section 3.1 for more details) [1,11,23,26,28].

An aberrant obturator artery (AOA) is an anatomical variation in which the OA often arises from the external iliac artery (EIA) (Figure 1) [3,21,27,28]. Select case studies have identified it in as many as 55.1% of individuals in their cohort [3,4,21,27,28]. Other alternative OA origins include the common iliac artery (CIA), inferior gluteal artery (IGA), internal pudendal artery (IPA), a common trunk for IGA and IPA, iliolumbar artery (ILA), EIA, a branch of the EIA, or by a dual root from both IIA and EIA sources [2,3,5,7,9,10,11,14,21,23,27,28,29,30,31,32,33]. 

Familiarity with the occurrence of an AOA is equally important for instructors teaching pelvic anatomy to students. Case studies highlighting this vascular variation provide anatomical instructors and surgeons with accurate information on how to identify such variants and their prevalence. In our studied population, the OA arose from the IIA bilaterally in ten of the eighteen individuals (55.6%). The OA branched from the posterior division unilaterally in two cadavers (11.1%); one on the left and one on the right. The individual with an OA originating from the left IIA posterior division had a right obturator artery (ROA) arising from the right IIA anterior division. The individual with an OA arising from the right IIA posterior division also had a left AOA (LAOA). Seven of the eighteen studied individuals (38.9%) had at least one AOA. Two cadavers had bilateral AOAs (11.1%). The cadavers analyzed were provided by the Maryland State Anatomy Board and the Uniformed Services University of the Health Sciences Donation Program.

A thorough understanding of the IIA branching patterns and their possible vascular variations is essential for obstetric surgeons, general surgeons, and interventional radiologists performing other types of pelvic procedures (i.e., hernia repairs or pelvic fractures) (see Section 3.4 for more details) [3,10,11,19,23,27,29,32,34].

## 2. Case Descriptions 

Case 1: A 96-year-old White Male with a listed cause of death of dementia exhibited a unilateral AOA on the left pelvic side with a Yamaki et al. (1998) Group B classification bilaterally and a Sañudo et al. (2011) Type B classification (Table 1 and Table 2) [8,26]. The left IIA posterior division branches (LIIA PDB) included the left iliolumbar artery (LILA), left superior gluteal artery (LSGA), left superior lateral sacral artery (LSLSA), left inferior lateral sacral artery (LILSA), and the left inferior gluteal artery (LIGA). The left IIA anterior division branches (LIIA ADB) were the left umbilical artery (LUA), with a branch of the left superior vesical artery (LSVA); the LSVA, with a branch of the left inferior vesical artery (LIVA); and the left prostatic artery (LPA), which has a common branch that bifurcated into the left internal pudendal artery (LIPA) and the left middle rectal artery (LMRA). The left aberrant obturator artery (LAOA) originated from the left deep inferior epigastric artery (LDIEA), which was a branch of the left external iliac artery (LEIA) (Figure 2). The right IIA posterior division branches (RIIA PDB) included the right iliolumbar artery (RILA), the right superior lateral sacral artery (RSLSA), the right inferior lateral sacral artery (RILSA), the right superior gluteal artery (RSGA), and the right inferior gluteal artery (RIGA). The right IIA anterior division branched (RIIA ADB) into the right umbilical artery (RUA), with a branch for the right superior vesical artery (RSVA); the right obturator artery (ROA); and a common branch for the right inferior vesical artery (RIVA), the right internal pudendal artery (RIPA), and the right middle rectal artery (RMRA). The LIIA PDB had a left accessory obturator artery branch (LAOAB) providing multiple arterial branches to the left iliacus and iliopsoas muscles. The ROA provided multiple arterial branches to these muscles on the right side. There was no distinct separation between the anterior and posterior divisions on either side of the pelvis. 

Case 2: A 70-year-old White Male with a listed cause of death of cardiac arrest exhibited bilateral AOAs with a Yamaki et al. (1998) Group B classification on the left, a Group A classification on the right, and a Sañudo et al. (2011) Type B classification bilaterally (Table 1 and Table 2) [8,26]. The LIIA PDB included the LILA and the LSGA with a terminating branch of LIGA. The LIIA ADB were the LUA, a possible LSVA, a possible LIVA, and the LIPA (Figure 3). The RIIA PDB were the RLSLA and the RSGA. The RIIA ADB were the RUA, with a branch of RSVA; the right middle vesical (or vesiculodeferential) artery (RMVA); the RIVA; and the RIGA with a branch of the RMRA and the RIPA (Figure 4). There was a LAOA and a right aberrant obturator artery (RAOA) with this individual in which both originated from their respective DIEAs. Due to damage that occurred during student dissection, the LSLSA, LILSA, LMRA, RILA, and the RILSA were not identifiable and several branches could not be named with certainty.

Case 3: An 84-year-old White Female with a listed cause of death of senile degeneration of brain/Parkinson’s disease was found to have a unilateral AOA on the left pelvic side with a Yamaki et al. (1998) Group A classification bilaterally and a Sañudo et al. (2011) Type B classification (Table 1 and Table 2) [8,26]. The LIIA PDB were the LILA, the LSLSA, the LILSA, and the LSGA. The LIIA ADB included the LUA; the LSVA, with a branch of LUTA; the LIPA, with the left vaginal artery (LVA) branch and the LMRA branch; and the LIGA. The LAOA originated from the LDIEA (Figure 5). The RIIA PDB were the RILA, the RLSA, and the RSGA. The RIIA ADB were as follows: the RUA, with a RSVA branch that had a right uterine artery (RUTA) branch; the ROA; the RIPA, with a right vaginal artery (RVA) branch and a possible RMRA branch; and the RIGA. 

Case 4: A 91-year-old White Female with a listed cause of death of obstructive pulmonary disease was found to have a unilateral AOA on the left pelvic side with a Yamaki et al. (1998) Group B classification bilaterally and a Sañudo et al. (2011) Type B classification (Table 1 and Table 2) [8,26]. The LIIA PDB included the LILA; the LSLSA; the LILSA; and the LSGA, with a LIGA branch. The LIIA ADB were the LUA, with a LSVA branch and a possible LUTA branch; and the LIPA, with a possible LMRA branch. There was significant damage from student dissection resulting in the inability to discern positively all of the branches present, and no LVA was found. The LAOA originated from the LDIEA (Figure 6). The RIIA PDB were the RILA; the ROA, with several arterial branches to the iliacus and iliopsoas muscles; the RSLSA; the RILSA; and the RSGA. The RIIA ADB included the RUA, with a RSVA branch and a possible RUTA; the RIGA; and a common branch of RIPA possibly with a RMRA. Due to significant damage from student dissection, the RVA was not found and several branches could not be named with certainty (Figure 7).

Case 5: An 84-year-old White Female with a listed cause of death of acute hemorrhagic cerebrovascular accident (CVA) exhibited bilateral AOAs with a Yamaki et al. (1998) Group A classification on the left, Group B on the right, and a Sañudo et al. (2011) Type B classification bilaterally (Table 1 and Table 2) [8,26]. The LIIA PDB included the LILA, the LSLSA, the LILSA, and the LSGA. The LIIA ADB were the LUA, a common branch of LSVA and LUTA, the LVA, a common branch of LMRA and LIPA, and the LIGA. The LAOA originated from the LDIEA. A branch originating from the LAOA supplied the left obturator internus muscle. The LIIA PDB had a LAOAB providing multiple arterial branches to the left iliacus and iliopsoas muscles (Figure 8). The RIIA PDB included the RILA, a common branch of the RSLSA and the RILSA, the RSGA, and the RIGA. The RIIA ADB were the RUA, with a RSVA branch; the RUTA, with an RVA branch; and a common branch of the RMRA and the RIPA. There was also a right accessory obturator artery branch (RAOAB) from the RMRA that supplied the inferior aspect of the right obturator internus muscle. The RAOA originated from the RDIEA. There were two small branches originating from the RAOA that supply the superior aspect of the right obturator internus muscle (Figure 9).

Case 6: A 36 year-old White Female with a listed cause of death of metastatic breast cancer exhibited a unilateral AOA on the right pelvic side with a Yamaki et al. (1998) Group A classification bilaterally and a Sañudo et al. (2011) Type E classification (Table 1 and Table 2) [8,26]. The LIIA PDB were a common branch of the LILA and the LSLSA, the LILSA, and the LSGA. The LIIA ADB included the LUA; the LSVA; the LUTA; the LOA; an accessory LILSA (ALILSA); a common trunk of the LMRA and the LVA; and the LIGA, with a LIPA branch. The LOA provided multiple branches supplying the obturator internus, iliopsoas, and iliacus muscles. There was no distinct separation between the anterior and posterior division on the left pelvic side (Figure 10). The RIIA PDB were the RILA, the RSLSA, the RILSA, and the RSGA. The RIIA ADB included the RUA, with a RSVA branch; the RUTA; a common trunk of the RVA and the RMRA; and the RIGA, with a RIPA branch. The RAOA did not originate from the RDIEA, but rather, as an independent branch from the REIA (Figure 11).

Case 7: A 63-year-old White Male with a listed cause of death of renal cancer was found to have a unilateral AOA on the right pelvic side with a Yamaki et al. (1998) Group A classification bilaterally and a Sañudo et al. (2011) Type B classification bilaterally (Table 1 and Table 2) [8,26]. The LIIA PDB included the LILA, the LSLSA, the LILSA, and the LSGA. The LIIA ASB were the LUA, with a branch of the LSVA; the LIVA; and a common trunk of the LOA, the LIPA, and the LIGA. The LIPA also had a common trunk with the LMRA. The RIIA PDB included the RILA, the RLSA, and the RSGA. The RIIA ASB were the RUA, with a RSVA branch; the RMVA; the RIVA; the RIPA, with a RMRA branch; and the RIGA. The RAOA originated from the RDIEA (Figure 12). 

## 3. Discussion

### 3.1. Obturator Artery

The abdominal aorta divides into the right and left common iliac arteries in the range of L3–L5, with the most common site being anterolateral to the left side of L4. Each CIA normally bifurcates into the EIA and IIA within the range of the L4–L5 disc and the mid-height of the S2 vertebra [11,13,27,28]. The EIA mainly supplies the lower limbs. The IIA usually descends posteriorly to the superior margin of the greater sciatic foramen where it divides into the posterior and anterior divisions. The posterior division passes back to the greater sciatic foramen, while the anterior division descends towards the ischial spine. These divisions provide many of the branches that supply the pelvic viscera, pelvic walls, perineum, and the gluteal region. These visceral and parietal arteries branch in numerous ways, and variations are common [1,11,23,26,28]. In the male, there are normally four visceral branches: the superior vesical artery (SVA), inferior vesical artery (IVA), middle rectal (or hemorrhoidal) artery (MRA), and the IPA. There is also a smaller branch called the prostatic artery, which is usually a branch of the IVA. In the female, there are normally five visceral branches: the SVA, uterine artery (UTA), vaginal artery (VA), MRA, and the IPA. In both males and females, there are normally six parietal branches: the ILA, superior lateral sacral artery (SLSA), inferior lateral sacral artery (ILSA), superior gluteal artery (SGA), IGA, and the OA. After birth, the umbilical artery (UA) becomes a ligament. The middle vesical (or vesiculodeferential) artery (MVA) usually derives from or is an adjacent branch to the IVA in males and the UTA or VA in females. 

Yamaki et al. (1998) created a classification system (Groups A-D) for IIA branching patterns that was adapted from the Adachi (1928) classification method (Table 1) [1,26]. With the Yamaki et al. method, IIA branching patterns are classified based on the following three main branches: the SGA, IGA, and the IPA. The obturator artery is not used in the classification because of its high rate of origin variability from the IIA and EIA (Table 1). Group A is considered the basic IIA branching pattern because, of the identified cases, it occurs most frequently (60–80%). Group B is the second most frequent IIA branching pattern (15–30% of the population). Group C has been found in 5–7% of pelvic sides. Group D, a very rare pattern classification, has only been identified in one pelvic side (0.2%) [1,26]. 

The OA is a parietal extrapelvic branch of the IIA that usually arises from its anterior division (21–88.9% occurrence rate) with the obturator vein (OV) draining into the internal iliac vein (IIV) [3,5,7,8,9,10,11,12,13,14,21,23,27,28,29,30,31,32,34,35,36,37,38,39]. The OA usually arises either on the lateral or dorsolateral surface of the anterior division. It can, however, vary in its origin (6.6–63.63%) and has the greatest frequency of variation among the IIA branches [2,3,4,5,6,7,8,9,10,11,13]. In both males and females, the OA can arise from the CIA, the IGA (2–9%), the IPA (2–3.8%), a common trunk for the IGA and the IPA (10%), the ILA (1–3.33%), the EIA (1.1–4%), a branch of the EIA (8–33.3%), or by a dual root from both the IIA and the EIA sources (6.5%) [2,3,5,7,9,10,11,14,21,23,27,28,29,30,31,32,33]. The OA may also originate from the posterior division of the IIA (0.5–14.5%), usually as a branch of the SGA (2–16.1%) [2,7,9,10,11,12,21,23,28,30,31,32,33,34,37,38,40]. Additionally, the OA can have varying origins on the left and right side of the same pelvis [2,29]. In the studied population, the OA arose from the anterior division of the IIA bilaterally in ten of the eighteen individuals (55.6%). The OA branched from the posterior division unilaterally in two cadavers (11.1%), one on the left pelvic side and one on the right (Figure 7). The individual with a LOA originating from the left IIA posterior division also had a ROA arising from the IIA anterior division. The individual with a ROA arising from the right IIA posterior division also exhibited a LAOA (Figure 6). The OA may also have two (dual) or three origins (1–25%), and it is possible to have an accessory OA (30–40%) [5,9,22,25,29,38,39,41]. Two cadavers exhibited instances of accessory OAs, with one individual exhibiting it bilaterally (Figure 2, Figure 8 and Figure 9).

Sañudo et al. (2011) classifies the OA variations into six different types (A-F) (Table 2). Types A and B are the most common (35.5% and 22.5%, respectively) [8,29]. Type E is the second rarest (1.7% of cases). Pick et al. (1942) and Leite et al. (2017) are the only two articles known by the authors to have referenced Type F, the rarest OA variation type (1.66%), with one case described in each article [7,8,32].

After originating from the IIA, the OA traverses the lateral wall of the pelvis, inferior to the brim and enters the obturator foramen near its superior edge. Its usual course is medial to the obturator fascia; lateral to the ureter, ductus deferens, and peritoneum; and inferior to the obturator nerve. As the OA passes through the pelvis, it can provide several muscular and visceral branches to the iliac, vesical, and pubic regions [3,9,10,14,21,22,27,28,29,32,34,37,38]. The muscular branches can supply iliacus, iliopsoas, and obturator internus muscles. The OA also has branches that supply the iliac fossa. Prior to leaving the pelvic cavity, the OA can provide a pubic branch as a collateral circulation with the EIA system via the DIEA. It then passes from the pelvic cavity through the obturator canal to the medial compartment of the thigh. When leaving the pelvic cavity through the obturator foramen, the OA divides into two branches, the anterior and posterior branches that anastomose with the internal circumflex artery. The anterior branch supplies obturator externus, pectineus, adductor longus, adductor brevis, adductor magnus, and gracilis muscles. It terminates by anastomosing with the posterior branch of the obturator and medial circumflex femoral arteries. The posterior branch provides blood supply to the semimembranosus, semitendinosus, long head of biceps femoris, and the adductor magnus muscles. It then anastomoses with the IGA [3,11,21,27,28,29,32,42]. 

### 3.2. Aberrant Obturator Artery

An AOA is an anatomical variation in which the OA, a standard branch of the anterior division of the IIA, arises instead from the EIA. Previous studies have reported a frequency of as many as 55.1% of individuals [3,4,21,27,28]. It can arise from the DIEA (2.6–44%); directly from the EIA (1.1–10%); or from the femoral artery (FA) (1.1–1.66%) with the OV draining into the deep inferior epigastric vein (DIEV), the external iliac vein (EIV), or into the femoral vein (FV) [2,3,5,7,8,12,13,16,19,21,22,23,25,27,28,29,31,32,33,34,36,37,38,43,44,45,46,47,48,49,50,51]. Many studies have found an AOA origin from the DIEA to be more common in females than males. It is rarely found bilaterally [3,4,27,35,37]. Pai et al. 2009, however, reported a higher incidence of AOA origin from the DIEA in males (47%) than in females (26%) [5]. The AOA is also more frequently observed on the left rather than the right side of the pelvis [3,7]. The AOA then passes anteromedially to the external iliac vein, encircles the internal end of the femoral canal, passes over the pectineal ligament and descends towards the obturator foramen [14,24]. In the studied population, six of the seven individuals with unilateral or bilateral AOAs originated from the DIEA (33.3%) and one individual’s AOA originated directly from the EIA (5.6%) (Figure 11). Of the six individuals with an AOA originating from the DIEA, three were male and three were female. The individual that had an AOA originating directly from the EIA was female. Of the seven AOA cases, five had LAOAs (27.8%) and four had RAOAs (22.2%). 

### 3.3. Embryonic Development

During the fourth week of fetal life, the right and left dorsal aortae fuse caudal to the tenth dorsal intersegmental artery to form the descending aorta. The UA is the specialized paired ventral segmental branch passing through the connecting stalk, on each side. During the fifth week, the proximal part of each UA anastomoses with fifth dorsal lumbar intersegmental artery to form a new stem. This stem forms the dorsal root of UA, while the original ventral root of the UA degenerates. The dorsal root of the UA gives rise to two arterial plexuses (the abdominal and the pelvic). The pelvic plexus persists as the CIA and gives off branches that become the EIA and the IIA [8,21,29,32,34,40,42]. 

It appears that the OA forms as a result of uneven growth of an anastomosis of the EIA and IIA. In addition, the OA arises comparatively late to supply the medial side of the thigh. Both the uneven anastomosis growth and its late appearance may explain the occurrence of OA origin and trajectory variations [5,8,9,11,21,22,23,25,28,29,34,39,40,52]. In relation to the uneven anastomosis growth theory, some OA variations, such as its origin from the IIA posterior division, can be explained as vascular channels for anastomosis that persist in the posterior IIA division and those predestined for the OA may have disappeared or degenerated in the anterior division [5,11,13,25,27,28,40,50,53]. The AOA originating from the DIEA may be due to the underdevelopment or obliteration of a normal obturator at its origin and an enlargement of an anastomosis between the pubic branches of DIEA and OA behind the pubic body [35,47]. A dual origin of the OA may be interpreted as the presence of two source channels for the blood flow, one from the IIA and the other from the DIEA [5,50]. 

It should also be noted that OA origin variations may also occur later in life due to pathological conditions. These may involve venoclusive or arterial thromboembolic phenomenon and trauma or surgery in the pelvic area [12].

### 3.4. Clinical Significance

Accidental hemorrhage is the leading cause of obstetrical mortality in the United States of America. It is also the leading cause of maternal deaths in the developing world. Thus, a thorough understanding of the IIA branching patterns and their possible vascular variations is essential for obstetric surgeons [10,11,34,54]. Such knowledge is also crucially important for general surgeons and interventional radiologists performing other types of pelvic procedures (i.e., hernia repairs or pelvic fractures), as well as for anatomists teaching pelvic vasculature [3,19,23,27,29,32,38,39,50,51,55,56,57]. 

A vascular variant of the AOA that can be encountered in pelvic procedures is known as the corona mortis (CMOR). CMOR, meaning “the crown of death”, involves vascular communication(s) between the OA and EIA or DIEA vessels that is present in 8.22–84% of patients [1,5,10,12,13,14,18,19,22,24,29,32,38,39,41,46,50,51,56,57,58,59]. The wide variation in CMOR incidence suggests there are ethnic or regional differences [19]. This is an arterial branch variation that usually originates from the EIA, DIEA, or coexists with the OA and anastomoses with it, creating an arc around the internal end of the femoral canal above the superior pubic ramus [1,14,15,29,39,57,58]. It can be unilateral or bilateral, and there seems to be no significant difference in its incidence between males and females [46,60,61,62,63,64]. This vascular variant earns its name due to the significant risk of death raised by its injury, which can lead to substantial hemorrhage and difficult hemostasis [10,11,14,15,16,17,18,19,20,32]. During open or laparoscopic hernia surgery, unrecognized injury to this vessel can lead to significant hemorrhage into the extraperitoneal space between parietal peritoneum and transversalis fascia. The typical course of this anomalous vessel encircling the superior end of the femoral canal (thus “crown”) puts it at risk in open or laparoscopic hernia surgery. This is important given the current practice of sending patients home on the day of surgery. Unobserved hemorrhage especially in the elderly with compromised coronary circulation can be life threatening. Kashyap et al. (2019) and Sanna et al. (2018) found that the venous CMOR is much more prevalent than arterial CMOR and the majority of cases involved small caliber vessels (<4 mm). Anatomical case reports have identified a wide variety in the pattern and number of arterial or venous CMORs, and most cases demonstrated dissimilarity between left and right pelvic sides [12,18,19,24,38,51,59].

CMOR poses a risk in surgical procedures involving the inferior part of the anterior abdominal wall because the associated vessels run above and behind the superior pubic ramus in a relatively vertical direction [14,15,24]. These retropubic vessels are of paramount importance for surgeons treating pelvic and acetabular trauma, totally extraperitoneal (TEP) inguinal hernioplasties (especially during mesh fixation onto Cooper’s ligament), herniorraphies, transcatheter embolizations, muscle graft surgeries, lymphadenectomies, catheterizations, and during IIA aneurysm repairs [3,5,8,9,10,11,12,13,14,18,19,23,27,28,29,32,33,34,37,46,55,59].

Dissection near the superior pubic ramus and Bogros space during surgical intervention must be conducted cautiously with advanced knowledge of such vascular variations [5,9,11,12,16,18,22,23,50,51]. Prior to any pelvic procedure, preoperative angiographic analysis of bilateral internal and external iliac systems should be conducted to ensure adequate evaluation of a potential collateral supply [19,46]. The presence of an AOA is not exclusively a risk factor for surgical complications, but it could also be beneficial if the IIA and its collateral blood supply were to be ligated or obstructed. The AOA and its branches could be a source of collateral circulation, especially the branch to the femoral head [5,9,14,23,29,32,50]. Knowing the vascular pattern and awareness to assess for possible OA variations can decrease the risk of iatrogenic injury and may modify the surgical and procedural approaches to minimize the postsurgical complications [3,25,27,28,29,34,46,55,59]. 

## 4. Conclusions

OA origin variations, such as AOAs, are common in the literature and frequent in occurrence. Proficient knowledge of pelvic vascular anatomy is essential for performing embolizations, revascularization procedures, treating pelvic fractures, laparoscopic herniorrhaphies, and obstetrical procedures. Performing preoperative angiographic analysis to know the pelvic vascular pattern and having the awareness to assess for possible OA variations can decrease the risk of iatrogenic injury. It may also modify the surgical procedures to minimize the postsurgical complications. Such familiarity is equally important for anatomy instructors to convey such information to their students on the presence and frequency of such vascular variations.

## Figures and Tables

**Figure 1 diagnostics-10-00546-f001:**
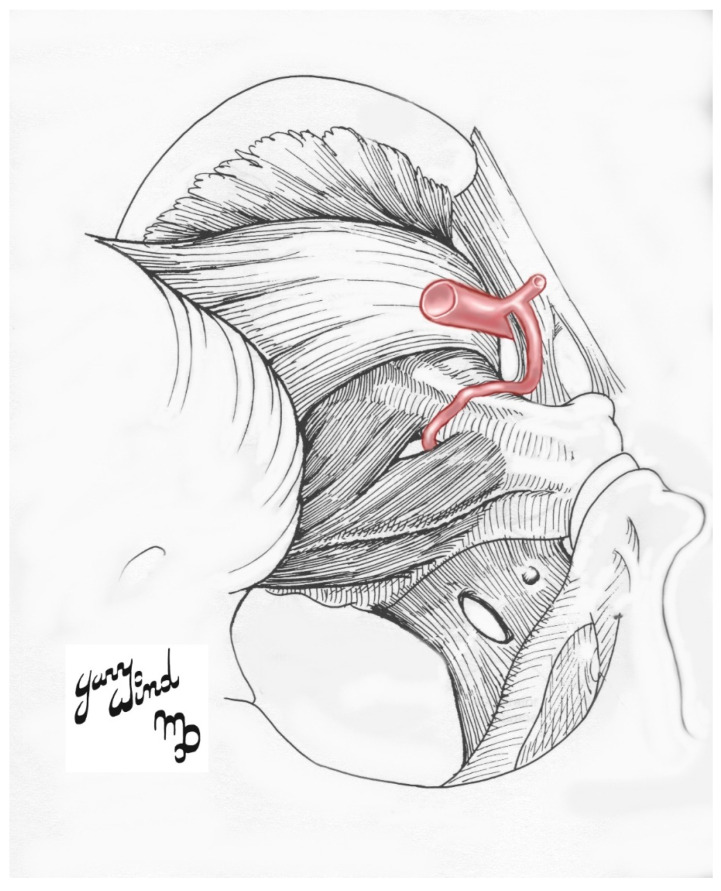
Illustrative schematic of the most common type of the aberrant obturator artery.

**Figure 2 diagnostics-10-00546-f002:**
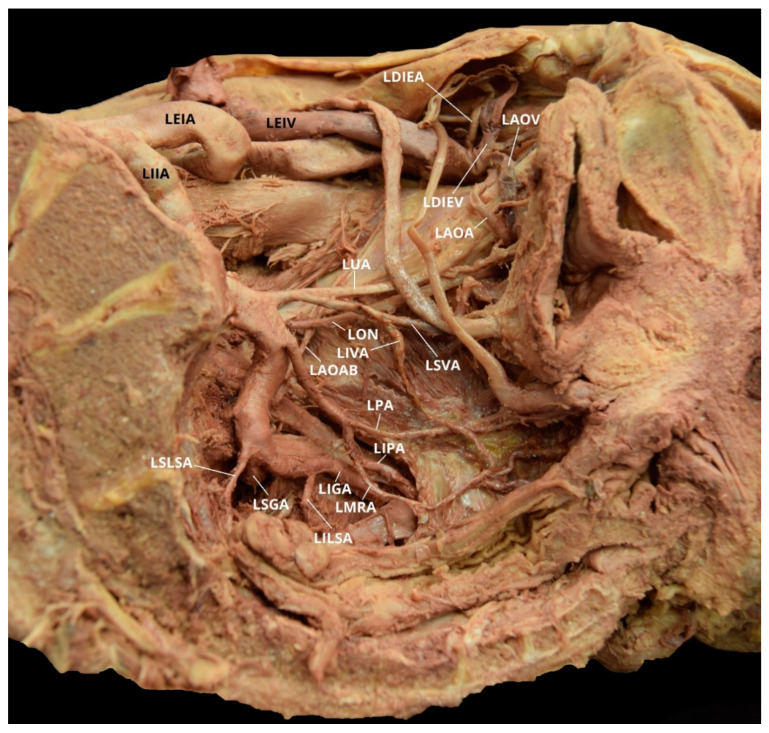
Facilitated display highlighting the left internal iliac artery branches with the left aberrant obturator artery for Case 1. LAOA = Left Aberrant Obturator Artery; LAOAB = Left Accessory Obturator Artery Branch; LAOV = Left Aberrant Obturator Vein; LDIEA = Left Deep Inferior Epigastric Artery; LDIEV = Left Deep Inferior Epigastric Vein; LEIA = Left External Iliac Artery; LEIV = Left External Iliac Vein; LIGA = Left Inferior Gluteal Artery; LIIA = Left Internal Iliac Artery; LILSA = Left Inferior Lateral Sacral Artery; LIPA = Left Internal Pudendal Artery; LIVA = Left Inferior Vesical Artery; LMRA = Left Middle Rectal Artery; LON = Left Obturator Nerve; LPA = Left Prostatic Artery; LSGA = Left Superior Gluteal Artery; LSLSA = Left Superior Lateral Sacral Artery; LSVA = Left Superior Vesical Artery; LUA = Left Umbilical Artery. The Left Iliolumbar Artery (LILA) is not pictured.

**Figure 3 diagnostics-10-00546-f003:**
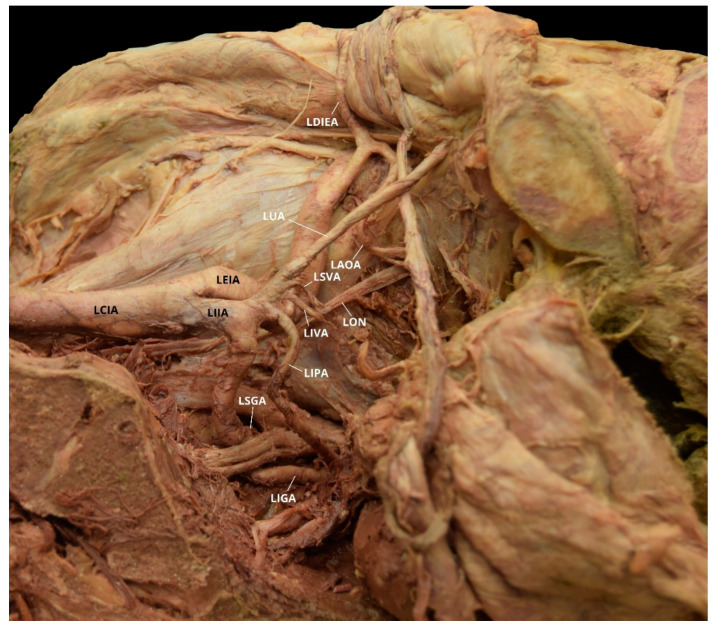
Facilitated display highlighting the left internal iliac artery branches with the left aberrant obturator artery for Case 2. LAOA = Left Aberrant Obturator Artery; LCIA = Left Common Iliac Artery; LDIEA = Left Deep Inferior Epigastric Artery; LEIA = Left External Iliac Artery; LIGA = Left Inferior Gluteal Artery; LIIA = Left Internal Iliac Artery; LIPA = Left Internal Pudendal Artery; LIVA = Left Inferior Vesical Artery; LON = Left Obturator Nerve; LSGA = Left Superior Gluteal Artery; LSVA = Left Superior Vesical Artery; LUA = Left Umbilical Artery. The Left Iliolumbar Artery (LILA), Left Superior Lateral Sacral Artery (LSLSA), Left Inferior Lateral Sacral Artery (LILSA), and the Left Middle Rectal Artery (LMRA) are not pictured.

**Figure 4 diagnostics-10-00546-f004:**
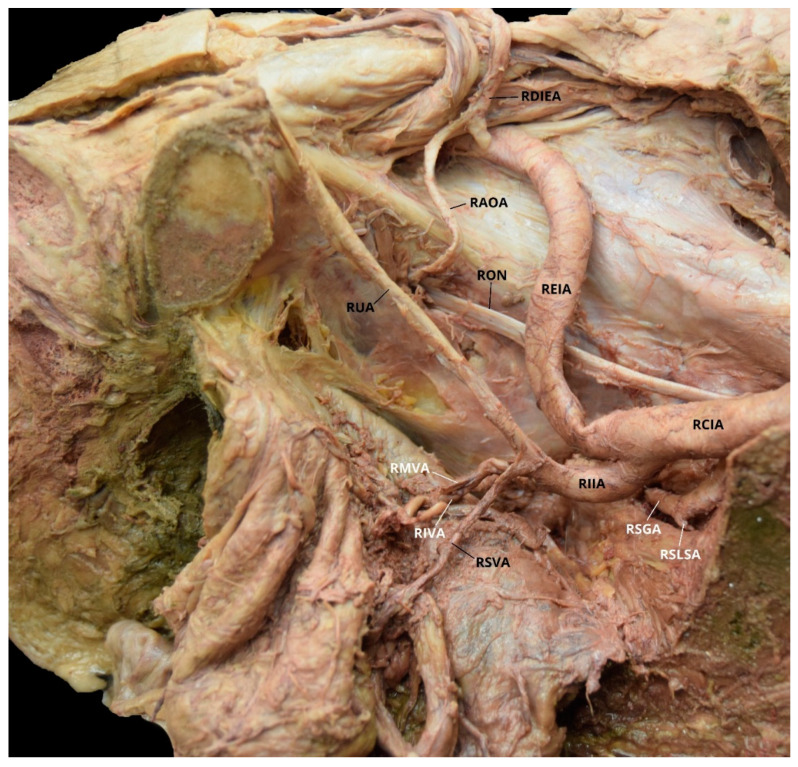
Facilitated display highlighting the right internal iliac artery branches with the right aberrant obturator artery for Case 2. RAOA = Right Aberrant Obturator Artery; RCIA = Right Common Iliac Artery; RDIEA = Right Deep Inferior Epigastric Artery; REIA = Right External Iliac Artery; RIIA = Right Internal Iliac Artery; RIVA = Right Inferior Vesical Artery; RON = Right Obturator Nerve; RMVA = Right Middle Vesical Artery; RSGA = Right Superior Gluteal Artery; RSLSA = Right Superior Lateral Sacral Artery; RSVA = Right Superior Vesical Artery; RUA = Right Umbilical Artery. The Right Inferior Gluteal Artery (RIGA), Right Iliolumbar Artery (RILA), Right Inferior Lateral Sacral Artery (RILSA), Right Internal Pudendal Artery (RIPA), and the Right Middle Rectal Artery (RMRA) are not pictured.

**Figure 5 diagnostics-10-00546-f005:**
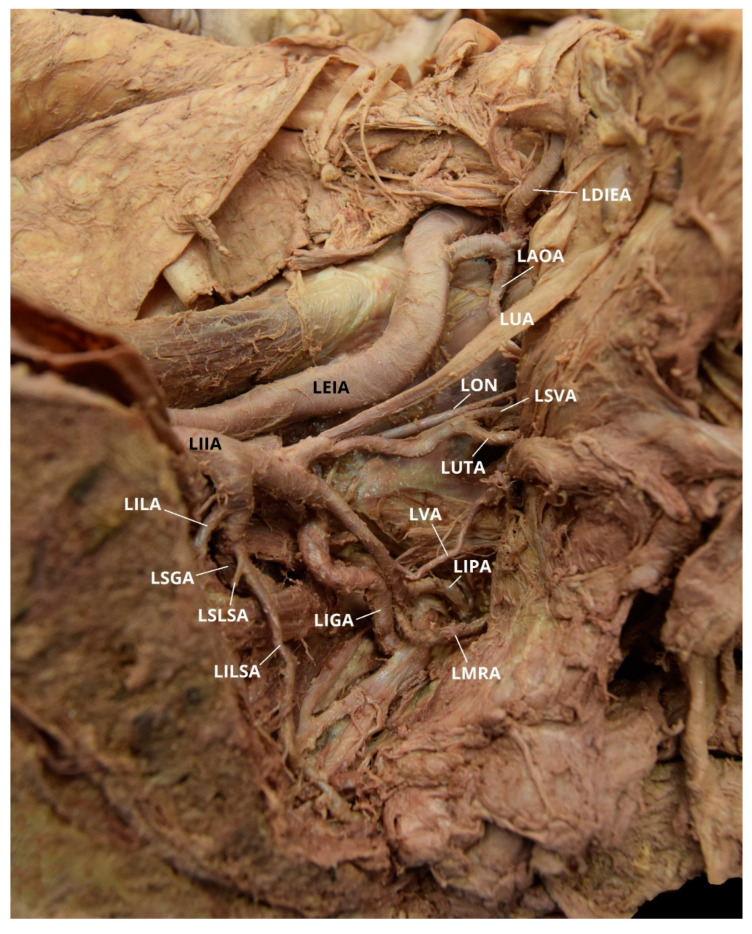
Facilitated display highlighting the left internal iliac artery branches with the left aberrant obturator artery for Case 3. LAOA = Left Aberrant Obturator Artery; LDIEA = Left Deep Inferior Epigastric Artery; LEIA = Left External Iliac Artery; LIGA = Left Inferior Gluteal Artery; LIIA = Left Internal Iliac Artery; LILA = Left Iliolumbar Artery; LILSA = Left Inferior Lateral Sacral Artery; LIPA = Left Internal Pudendal Artery; LMRA = Left Middle Rectal Artery; LON = Left Obturator Nerve; LSGA = Left Superior Gluteal Artery; LSLSA = Left Superior Lateral Sacral Artery; LSVA = Left Superior Vesical Artery; LUA = Left Umbilical Artery; LUTA = Left Uterine Artery; LVA = Left Vaginal Artery.

**Figure 6 diagnostics-10-00546-f006:**
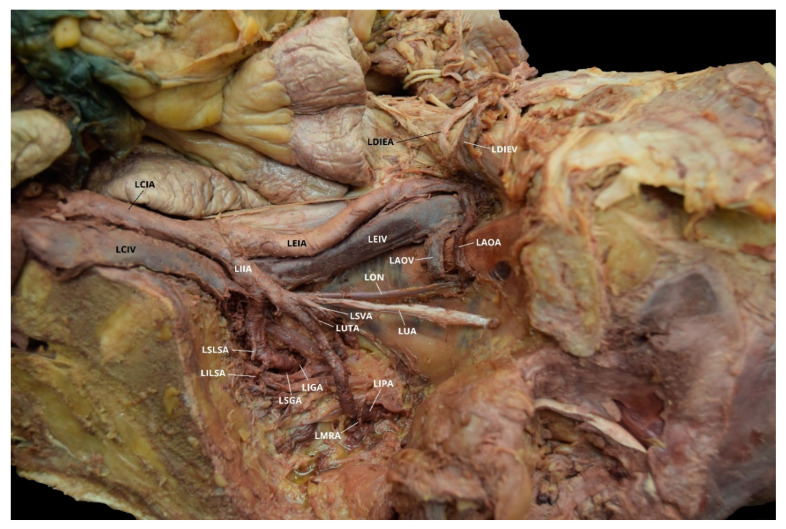
Facilitated display highlighting the left internal iliac artery branches with the left aberrant obturator artery for Case 4. LAOA = Left Aberrant Obturator Artery; LAOV = Left Aberrant Obturator Vein; LCIA = Left Common Iliac Artery; LCIV = Left Common Iliac Vein; LDIEA = Left Deep Inferior Epigastric Artery; LDIEV = Left Deep Inferior Epigastric Vein; LEIA = Left External Iliac Artery; LEIV = Left External Iliac Vein; LIGA = Left Inferior Gluteal Artery; LIIA = Left Internal Iliac Artery; LILSA = Left Inferior Lateral Sacral Artery; LIPA = Left Internal Pudendal Artery; LMRA = Left Middle Rectal Artery; LON = Left Obturator Nerve; LSGA = Left Superior Gluteal Artery; LSLSA = Left Superior Lateral Sacral Artery; LSVA = Left Superior Vesical Artery; LUA = Left Umbilical Artery; LUTA = Left Uterine Artery. The Left Iliolumbar Artery (LILA) and the Left Vaginal Artery are not pictured.

**Figure 7 diagnostics-10-00546-f007:**
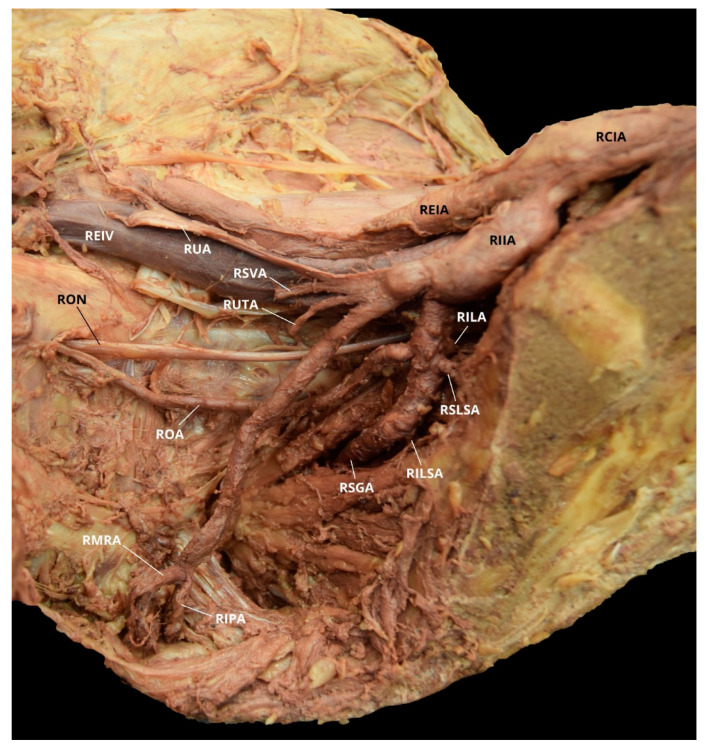
Facilitated display highlighting the right internal iliac artery branches for Case 4. RCIA = Right Common Iliac Artery; REIA = Right External Iliac Artery; REIV = Right External Iliac Vein; RIIA = Right Internal Iliac Artery; RILA = Right Iliolumbar Artery; RILSA = Right Inferior Lateral Sacral Artery; RIPA = Right Internal Pudendal Artery; RMRA = Right Middle Rectal Artery; ROA = Right Obturator Artery; RON = Right Obturator Nerve; RSGA = Right Superior Gluteal Artery; RSLSA = Right Superior Lateral Sacral Artery; RSVA = Right Superior Vesical Artery; RUA = Right Umbilical Artery; RUTA = Right Uterine Artery. The Right Inferior Gluteal Artery (RIGA) and Right Vaginal Artery (RVA) are not pictured.

**Figure 8 diagnostics-10-00546-f008:**
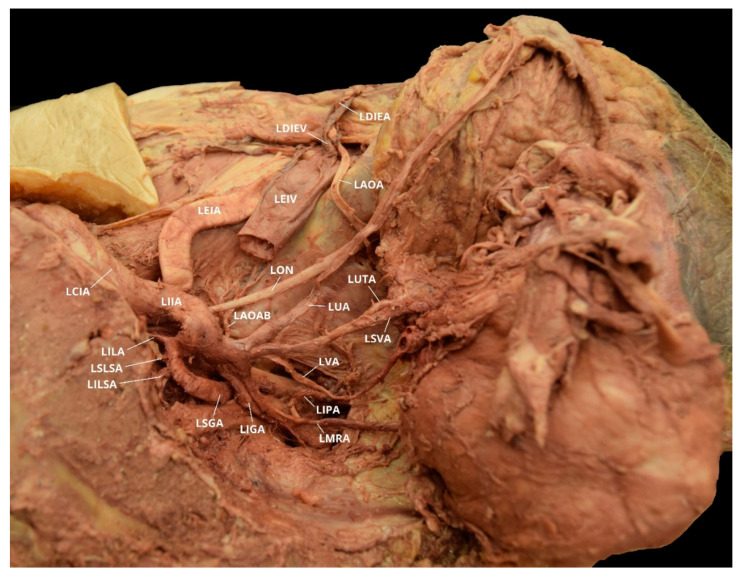
Facilitated display highlighting the left internal iliac artery branches with the left aberrant obturator artery for Case 5. LAOA = Left Aberrant Obturator Artery; LAOAB = Left Accessory Obturator Artery Branch; LCIA = Left Common Iliac Artery; LDIEA = Left Deep Inferior Epigastric Artery; LDIEV = Left Deep Inferior Epigastric Vein; LEIA = Left External Iliac Artery; LEIV = Left External Iliac Vein; LIGA = Left Inferior Gluteal Artery; LIIA = Left Internal Iliac Artery; LILA = Left Iliolumbar Artery; LILSA = Left Inferior Lateral Sacral Artery; LIPA = Left Internal Pudendal Artery; LMRA = Left Middle Rectal Artery; LSVA = Left Superior Vesical Artery; LON = Left Obturator Nerve; LSGA = Left Superior Gluteal Artery; LSLSA = Left Superior Lateral Sacral Artery; LUA = Left Umbilical Artery; LUTA = Left Uterine Artery; LVA = Left Vaginal Artery.

**Figure 9 diagnostics-10-00546-f009:**
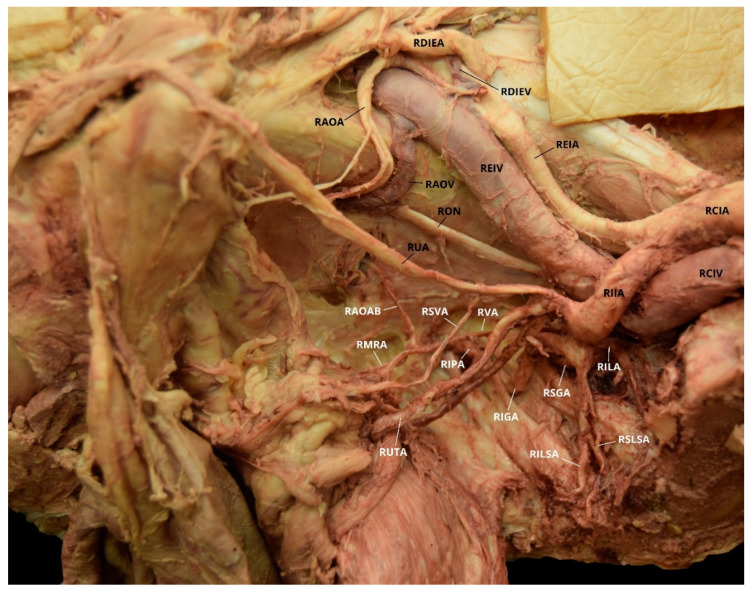
Facilitated display highlighting the right internal iliac artery branches with the right aberrant obturator artery for Case 5. RAOA = Right Aberrant Obturator Artery; RAOAB = Right Accessory Obturator Artery Branch; RCIA = Right Common Iliac Artery; RCIV = Right Common Iliac Vein; RDIEA = Right Deep Inferior Epigastric Artery; RDIEV = Right Deep Inferior Epigastric Vein; REIA = Right External Iliac Artery; REIV = Right External Iliac Vein; RIGA = Right Inferior Gluteal Artery; RIIA = Right Internal Iliac Artery; RILA = Right Iliolumbar Artery; RILSA = Right Inferior Lateral Sacral Artery; RIPA = Right Internal Pudendal Artery; RMRA = Right Middle Rectal Artery; ROA = Right Obturator Artery; RON = Right Obturator Nerve; RSGA = Right Superior Gluteal Artery; RSLSA = Right Superior Lateral Sacral Artery; RSVA = Right Superior Vesical Artery; RUA = Right Umbilical Artery; RUTA = Right Uterine Artery; RVA = Right Vaginal Artery.

**Figure 10 diagnostics-10-00546-f010:**
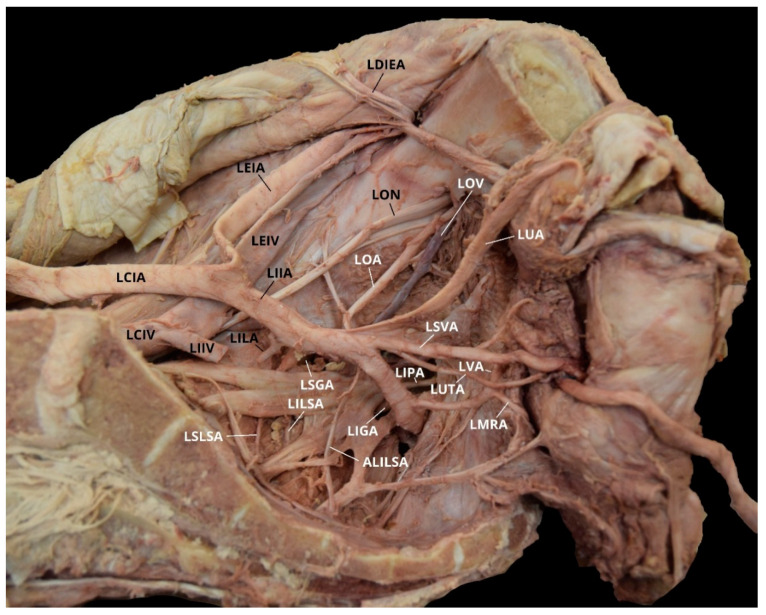
Facilitated display highlighting the left internal iliac artery branches for Case 6. ALILSA = Accessory Left Inferior Lateral Sacral Artery; LCIA = Left Common Iliac Artery; LCIV = Left Common Iliac Vein; LDIEA = Left Deep Inferior Epigastric Artery; LEIA = Left External Iliac Artery; LEIV = Left External Iliac Vein; LIIA = Left Internal Iliac Artery; LIGA = Left Inferior Gluteal Artery; LIIV = Left Internal Iliac Vein; LILA = Left Iliolumbar Artery; LILSA = Left Inferior Lateral Sacral Artery; LIPA = Left Internal Pudendal Artery; LMRA = Left Middle Rectal Artery; LOA = Left Obturator Artery; LON = Left Obturator Nerve; LOV = Left Obturator Vein; LSGA = Left Superior Gluteal Artery; LSLSA = Left Superior Lateral Sacral Artery; LSVA = Left Superior Vesical Artery; LUA = Left Umbilical Artery; LUTA = Left Uterine Artery; LVA = Left Vaginal Artery.

**Figure 11 diagnostics-10-00546-f011:**
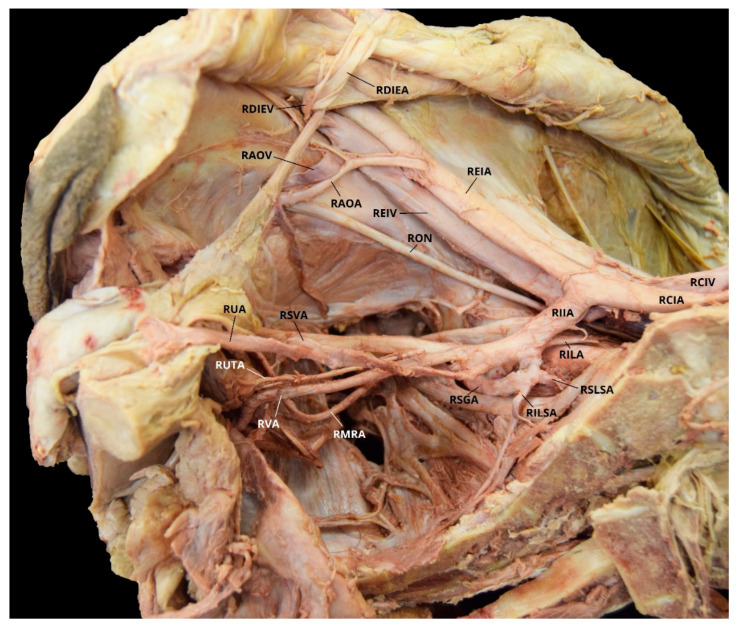
Facilitated display highlighting the right internal iliac artery branches with the right aberrant obturator artery for Case 6. RAOA = Right Aberrant Obturator Artery; RAOV = Right Aberrant Obturator Vein; RCIA = Right Common Iliac Artery; RCIV = Right Common Iliac Vein; RDIEA = Right Deep Inferior Epigastric Artery; RDIEV = Right Deep Inferior Epigastric Vein; REIA = Right External Iliac Artery; REIV = Right External Iliac Vein; RIIA = Right Internal Iliac Artery; RILA = Right Iliolumbar Artery; RILSA = Right Inferior Lateral Sacral Artery; RMRA = Right Middle Rectal Artery; RON = Right Obturator Nerve; RSGA = Right Superior Gluteal Artery; RSLSA = Right Superior Lateral Sacral Artery; RSVA = Right Superior Vesical Artery; RUA = Right Umbilical Artery; RUTA = Right Uterine Artery; RVA = Right Vaginal Artery. The Right Inferior Gluteal Artery (RIGA) and the Right Internal Pudendal Artery (RIPA) are not pictured.

**Figure 12 diagnostics-10-00546-f012:**
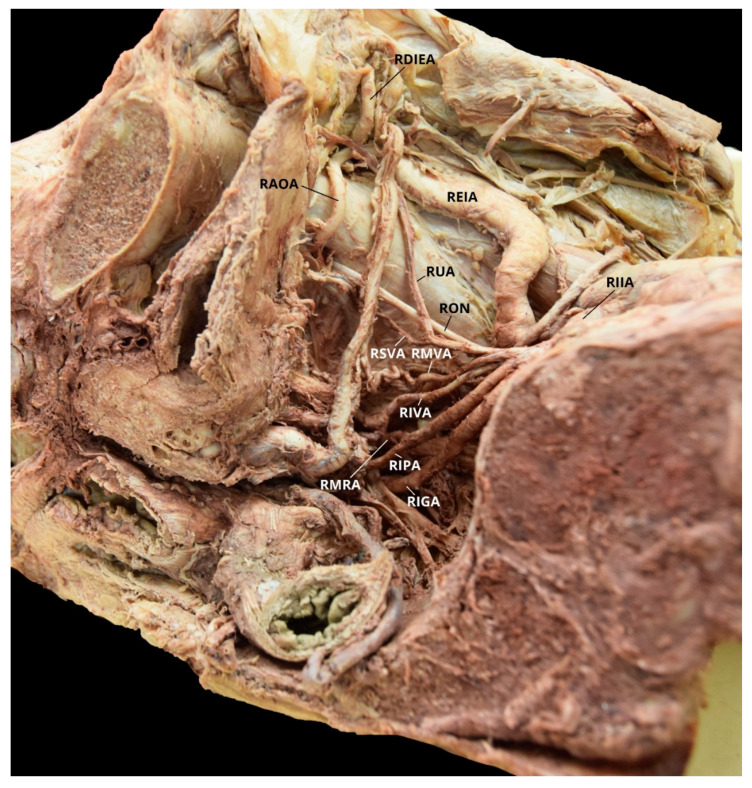
Facilitated display highlighting the right internal iliac artery branches with the right aberrant obturator artery for Case 7. RAOA = Right Aberrant Obturator Artery; RDIEA = Right Deep Inferior Epigastric Artery; REIA = Right External Iliac Artery; RIGA = Right Inferior Gluteal Artery; RIIA = Right Internal Iliac Artery; RIPA = Right Internal Pudendal Artery; RIVA = Right Inferior Vesical Artery; RMRA = Right Middle Rectal Artery; RMVA = Right Middle Vesical Artery; RON = Right Obturator Nerve; RSVA = Right Superior Vesical Artery; RUA = Right Umbilical Artery. The Right Iliolumbar Artery (RILA), Right Inferior Lateral Sacral Artery (RILSA), Right Superior Gluteal Artery (RSGA), and the Right Superior Lateral Sacral Artery (RSLSA) are not pictured.

**Table 1 diagnostics-10-00546-t001:** Yamaki et al. (1998) [26] IIA branching pattern classification system. SGA = superior gluteal artery; IGA = inferior gluteal artery; IPA = internal pudendal artery.

Group	Description of IIA Divisions
A	Two branches: (1) SGA and (2) a common trunk of IGA and IPA (common gluteal-pudendal trunk)
B	Two branches: (1) IPA and (2) a common gluteal trunk of SGA and IGA
C	Three branches: (1) SGA, (2) IGA, and (3) IPA
D	Two branches: (1) a common trunk of SGA and IPA and (2) IGA

**Table 2 diagnostics-10-00546-t002:** Sañudo et al. (2011) [8] obturator artery variation types. OA = obturator artery; DIEA = deep inferior epigastric artery; IIA = internal iliac artery; EIA = external iliac artery; FA = femoral artery.

Type	Description of OA Variation
A	The OA arises from anterior division of the IIA
B	The OA arises from the DIEA
C	The OA is a branch of the posterior division of the IIA
D	The OA arises from the IIA, above its final branching
E	The OA arises from the EIA
F	The OA arises from the FA

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
