# Peer review of "Frequency and Clinical Review of the Aberrant Obturator Artery: A Cadaveric Study"

_diagnostics, 2020, doi:10.3390/diagnostics10080546_

Round 1

Reviewer 1 Report

It appears that a so-called “aberrant” obturator artery is actually quite common. Perhaps you could comment on whether the term “aberrant” is accurate. Should the term be changed?

Table 1. Remove the word “the” from the lists of arteries. E.g., “the SGA” should be “SGA”.

The authors seem to make a point throughout the manuscript of right versus left arteries. For example, they refer to the aberrant arteries as Left aberrant obturator artery or right aberrant obturator artery. How much does the side of the arteries matter in this case? While it’s certainly relevant to note in the case studies on which side(s) the variants occurred, I wonder if indicating the side in each mention of each artery is necessary. For example, are the “L”s necessary prior to the name of each artery in Figure 3 etc? Or is there some developmental reason to make an important distinction between the sides?

Beautiful cadaveric figures with clear, excellent labeling.

In Case 5, define CVA before using the acronym.

The first paragraph of the Discussion should be moved to the Introduction. This information is important for setting up the context for the variations. Similarly, it would be worth adding to the Introduction a brief mention of the possible clinical significance of variation in these variants, such as some of the fascinating and relevant details included in 4.4 of the Discussion. This information would better set up the manuscript and provide the reader with better context going into the case studies.

Line 352, “can be explain” should be “can be explained”.

Author Response

Reviewer 1-

Thank you very much for your feedback. My responses are in red after your comments/suggestions.

It appears that a so-called “aberrant” obturator artery is actually quite common. Perhaps you could comment on whether the term “aberrant” is accurate. Should the term be changed?

The definition of "aberrant" is "departing from an accepted standard" or "diverging from the normal type"; it does not reference the prevalence of the variation using this terminology. We believe aberrant is still the proper term to use to describe this variation. 

Table 1. Remove the word “the” from the lists of arteries. E.g., “the SGA” should be “SGA”.

These changes have been made. Thank you for bringing it to our attention.

The authors seem to make a point throughout the manuscript of right versus left arteries. For example, they refer to the aberrant arteries as Left aberrant obturator artery or right aberrant obturator artery. How much does the side of the arteries matter in this case? While it’s certainly relevant to note in the case studies on which side(s) the variants occurred, I wonder if indicating the side in each mention of each artery is necessary. For example, are the “L”s necessary prior to the name of each artery in Figure 3 etc? Or is there some developmental reason to make an important distinction between the sides?

The reason for including left (L) and right (R) is due to the fact that the variation is found bilaterally in several of the cases so in order to describe these cases correctly, side needs to be included. To be consistent between cases, side has been included also in the unilateral cases.

Beautiful cadaveric figures with clear, excellent labeling.

Thank you so much!

In Case 5, define CVA before using the acronym.

The definition has been added before the acronym. Thank you for pointing this out!

The first paragraph of the Discussion should be moved to the Introduction. This information is important for setting up the context for the variations. Similarly, it would be worth adding to the Introduction a brief mention of the possible clinical significance of variation in these variants, such as some of the fascinating and relevant details included in 4.4 of the Discussion. This information would better set up the manuscript and provide the reader with better context going into the case studies.

What I've tried to do is provide a succinct condensation of the first paragraph of discussion 4.1 and 4.4 in the introduction. These summaries allow the reader to become more engage with the article by providing a transition platform.

Line 352, “can be explain” should be “can be explained”.

This change has been made, thanks!

Reviewer 2 Report

Accept as it is

Author Response

Thank you very much for reviewing and approving the article. I greatly appreciate you taking the time to review it!